# Clinical Profile of 24 AIDS Patients with Cryptococcal Meningitis in the HAART Era: A Report from an Infectious Diseases Tertiary Hospital in Western Romania

**DOI:** 10.3390/diagnostics12010054

**Published:** 2021-12-28

**Authors:** Iosif Marincu, Cosmin Citu, Iulia Vidican, Felix Bratosin, Mihai Mares, Oana Suciu, Stefan Frent, Adrian Vasile Bota, Madalina Timircan, Melania Lavinia Bratu, Mirela Loredana Grigoras

**Affiliations:** 1Methodological and Infectious Diseases Research Center, Department of Infectious Diseases, “Victor Babes” University of Medicine and Pharmacy, 300041 Timisoara, Romania; imarincu@umft.ro (I.M.); iulia.georgianabogdan@gmail.com (I.V.); felix.bratosin7@gmail.com (F.B.); mycomedica@gmail.com (M.M.); suciu.oana@umft.ro (O.S.); frentz.stefan@umft.ro (S.F.); bota.adrian1@yahoo.com (A.V.B.); timircan.madalina@yahoo.com (M.T.); bratu.lavinia@umft.ro (M.L.B.); grigoras.mirela@umft.ro (M.L.G.); 2Department of Psychology, “Victor Babes” University of Medicine and Pharmacy, 300041 Timisoara, Romania

**Keywords:** AIDS, *Cryptococcus neoformans*, HIV, amphotericin B, flucytosine, HAART, cryptococcal meningitis

## Abstract

Management of cryptococcal infections among patients suffering from acquired immunodeficiency syndrome (AIDS) represents a medical challenge. This retrospective study aims to describe the disease management and outcomes among 24 AIDS patients who suffered from *Cryptococcus neoformans* meningitis. The parameters evaluated from our patients’ database records include epidemiological data, clinical manifestations, biochemical and microbiological analysis of patients’ cerebrospinal fluid (CSF), treatment profiles, and disease outcomes. All patients included in the study had a lymphocyte count of less than 200 CD4/mm^3^. Of the 24 patients included in this study, five had been diagnosed with HIV infection since childhood, after receiving HIV-infected blood transfusions. The most prominent symptom was fatigue in 62.5% of patients, followed by nausea/vomiting and headache. Seven patients had liver cirrhosis due to hepatitis B virus (HBV) or hepatitis C virus (HCV) infection, while Kaposi sarcoma and cerebral toxoplasmosis were found in two patients. Six out of 24 patients died due to bacterial sepsis and acute respiratory distress syndrome (ARDS). High intracranial pressure was the strongest predictive factor for mortality (OR = 2.9), followed by ARDS (OR = 1.8), seizures at disease onset (OR = 1.4), and diabetes mellitus (OR = 1.2). Interestingly, patients younger than 40 years old had a significantly lower survival rate than that of the older patients. Before developing Cryptococcal meningitis, all patients had low adherence to the early ART treatment scheme and skipped the follow-up visits. All patients received a combination of amphotericin B and flucytosine as induction therapy, adding fluconazole for maintenance. Simultaneously, AIDS HAART was initiated at diagnosis of the cryptococcal infection. A combined regimen of antifungals and highly active antiretroviral therapy showed improved patient recovery with minor side effects.

## 1. Introduction

People living with human immunodeficiency virus (HIV) infection are prone to opportunistic infection with several microorganisms, including *Cryptococcus neoformans* yeast. This fungus has encapsulated morphology, causing a disease known as cryptococcal meningitis (CM) [1]. It has been estimated that despite the availability of antiretroviral treatments (ART), more than one million individuals, both immunocompromised and immunocompetent, suffer from the CM, leading to 625,000 deaths annually [2]. In Romania, a developing European country, the problem of HIV-positive adults is acute because a lot of people have been infected since childhood, before 1989, in the Communist era, by improper medical care and lack of screening for donated blood transfusions. Families wanted to protect them emotionally, and as such, modest diagnostic disclosure programs were developed later. In Europe, by 2018, 2,300,000 people were living with HIV, with ART being available to 54% (1,258,000) of them [3]. Regarding HIV/AIDS management in Europe, improvement for testing and diagnosis is needed because data from two large cohort studies (EuroSIDA and COHERE) released in 2020 report that around half of HIV diagnoses in Europe are late [4].

*Cryptococcus neoformans* yeast is ubiquitous in the environment, having a spherical-to-oval shape, measuring 5–10 µm in diameter with a polysaccharide carbohydrate capsule. Its morphology is considered the major virulence factor because it can negatively interfere with phagocytosis, leukocyte migration, and complement system activation. Other virulence factors include the production of oxidase and protease enzymes and optimal growth at 37 °C [5]; thus, *C. neoformans* can survive and thrive at host body temperature, essential for establishing infection in humans. The fungus does not produce toxins and sexually reproduces utilizing its two matings, type a and α. After budding, it shows hyphal morphology and divides by miosis into basidiospores. Worldwide, most cryptococcal infections in humans are caused by mating type MAT α and genotype A [6]. Based on the antigen detection test, two *Cryptococcus* varieties and five serotypes are identified as *C. neoformans* with three serotypes, A, D, and AD, and *C. gattii*, serotypes B and C. The two types have different geographical distributions, morphological characteristics, and pathogenicity, *C. neoformans* affecting most immunocompromised patients and *C. gattii* affecting healthy individuals.

*C. neoformans* is found ubiquitously in soil contaminated by bird excreta, especially pigeons. On the other hand, *C. gattii* is often found in regions such as Southeast Asia and Africa in tropical and subtropical climates, being associated with several species of eucalyptus trees [7]. The fungus enters the body via the respiratory tract and can generate pneumonia-like lesions or remain dormant, depending on the host reaction. According to the Center for Disease Control (CDC), most people breathe in the microscopic pathogen at some point in life but never develop symptoms [8]. By adulthood, many people develop anticryptococcal antibodies, suggesting previous contact with the fungus [9]. The fungus mainly affects five essential organs: the lungs, central nervous system, eyes, prostate, and skin. For people with different comorbidities, including AIDS, decompensated liver cirrhosis, chemotherapy, suppressive therapy for organ recipients, diabetes mellitus, lymphoma, leukemia, or other malignancies, the cell immunity involved in intracellular clearing of the fungus is impaired, leading to higher chances of symptomatic infection [10].

The mechanism involved in damage to host macrophage in cryptococcal infection involves phagolysosomal damage to membranes, organelle dysfunction, exocytosis, and cytoskeletal abnormalities. As described above, *C. neoformans* may be inhaled in the pulmonary macrophages but remain dormant, and later reactivation can occur if the host becomes immunocompromised. One unique feature of *C. neoformans* is replication and division inside macrophage phagolysosomes. The novel mechanism evaluated first in vitro and later confirmed in vivo revealed a pathogenic strategy involving the accumulation of intracellular polysaccharides in cytoplasmic vacuoles [11]. The importance of the interaction between *C. neoformans* and macrophages results from the latest studies highlighting the association between specific yeast behaviors and cryptococcosis outcomes under the same experimental conditions [12].

This study is focused on the clinical manifestations, treatment, and outcome of CM in patients with AIDS. The mortality from CM remains high (30–50%) [13,14] despite free access to ART and antifungal therapy, with the highest rates in developing countries, including sub-Saharan Africa and Southeast Asia [2]. Thus, considering the relatively low incidence of *Cryptococcus neoformans* meningitis on the European continent, this study aimed to describe the disease management and outcomes among 24 AIDS patients who suffered the disease and compare differences in outcomes between the group of survivors and the nonsurvivors.

## 2. Materials and Methods

We retrospectively analyzed the clinical manifestations, neuroimaging features, and laboratory reports of 24 adult patients diagnosed with cryptococcal meningitis admitted to the “Victor Babes” Hospital for Infectious Diseases and Pulmonology over a decade, from 1 January 2010 to 31 December 2020, using the hospital electronic database. All cases of *C. neoformans* meningitis were confirmed in our hospital by the cerebrospinal cultures, positive Ag titer, and India ink staining. Patient demographic data, HIV acquisition time, prior AIDS-defining conditions, comorbidities, symptomatology at admission, laboratory findings, CSF analysis, treatment side effects, comorbidities, and complications were included in our research. The therapeutic results of these patients were evaluated and recorded as the outcome on discharge. All patients gave their informed consent at the moment of admission. The study was conducted following the Declaration of Helsinki, and the protocol was approved by the Ethics Committee of Hospital of Infectious Diseases and Pulmonology, “Victor Babes” Timisoara.

AIDS was defined according to the CDC definition [10], HIV infection being classified as C3 stage when the CD4 cell count drops below 200 cells/mm^3^ or when patients develop certain opportunistic infections. CM was defined and diagnosed when *C. neoformans* was isolated in culture from cerebrospinal fluid, with positive CSF India ink staining in direct microscopy, or with the detection of a positive Ag titer. Patients who previously completed treatment for CM and again grew a CSF culture of *C. neoformans* were considered a relapse [15].

Data were analyzed with SPSS version 26. Descriptive statistics were calculated for quantitative and qualitative variables of the study. Mean and standard deviation were calculated for quantitative variables, and qualitative variables are presented as frequency and percentages. Associations between mortality and study variables were assessed. Normality was assessed using the Shapiro–Wilk test at a significance level > 0.05. Bivariate analysis was performed for qualitative variables. The Fisher exact test and chi-square test were used to determine differences in proportions. For quantitative variables, an independent *t*-test for normally distributed data and a Mann–Whitney U test for non-normally distributed data were used. A *p*-value < 0.05 was regarded as the significance threshold. Survival analysis was also performed. The odds ratio was calculated after performing a logistic regression analysis.

## 3. Results

A total of 24 patients with cryptococcal meningitis secondary to AIDS treated in the hospital were enrolled in this study. All patients had class C3 HIV and *C. neoformans* infection serotype A. Fifteen (62.5%) patients received the treatment regimen one, two (8.3%) of the patients received the regimen number two, three patients (12.5%) were treated according to scheme number three, while the other four (16.7%) were treated using scheme number four. All 24 patients were treated with amphotericin (Amp) B (0.5–1 mg/kg/day) and oral flucytosine (150 mg/kg/day) for two weeks as induction therapy, followed by oral flucytosine, 400 mg/day for ten weeks of treatment, mannitol solution 20%, 3 × 250 mL/day, and dexamethasone, 4 mg/mL, 4 × 4 mg/day (Table 1). One patient had a relapse after completing treatment for CM one year later.

The mean age of patients was 41 years old, ranging from 32 to 52 years. Out of 24 patients, 14 (58.3%) were older than 40 years. The proportion of males was higher than that of females (70.9% vs. 29.1%), and most of them were rural inhabitants (75.0%) (Table 2). It was known that 19 patients (79.2%) acquired HIV through sexual contact, although the proportion of heterosexual and homosexual intercourse remains unclear. The other five cases were known to have had an HIV infection since childhood, from the children’s HIV cohort. The mean HIV viral load of patients at the time of beginning AIDS antiretroviral treatment was 145,000 copies/µL.

The laboratory profile is shown in Table 3; WBC, Ly%, ESR, CRP, fibrinogen, Ht%, Hg, AST, ALT, and serum creatinine are presented in terms of mean and standard deviation. The CSF characteristics, including WBC, protein CSF, glucose CSF, lactate CSF, CSF opening pressure, CD4, CD8, CD3, and the ratio of CD4/CD8, are also described. Among these patients, six (25.0%) required ICU admission, for acute kidney injury in two patients and for ARDS in three patients, as described in Table 2. The mean hospitalization stay was 25.27 ± 11, ranging from 11 to 42 days. In-hospital complications included hepatic cytolysis syndrome, acute kidney injury, anemia, septic shock, oral candidiasis, nosocomial pneumonia, tuberculosis, and ARDS.

In Table 4, we describe the association between outcome (mortality) and other factors. Five out of six patients who died were younger than 40 years old. Interestingly, the survival rate was higher in older patients than in younger ones, and the difference between proportions was found to be statistically significant (*p*-value = 0.033). The mean viral load was compared between patients who died and those who survived (158 thousand copies/µL vs. 139 thousand copies/µL). However, there were no statistically significant differences between the proportions of survivors and nonsurvivors based on the AIDS treatment scheme received (*p*-value = 0.782).

Signs and symptoms at the time of presentation, as described in Figure 1 and Table 4 included fever in 12 patients (50.0%); 14 (58.3%) complained of nausea and vomiting, 13 (54.1%) had a headache; visual impairment was noticed in 10 (41.7%), 10 (41.7%) had an altered sensorium, and 9 patients (38.0%) described stiff neck complaints. Fatigue was the most common finding, affecting 15 patients (62.5%). Seven patients (29.1%) presented with diarrhea and night sweats, while eight (33.3%) suffered weight loss. Focal deficit and generalized lymphadenopathy were found in four patients (16.7%), and five patients (20.8%) had seizures at disease onset. The least common finding was ARDS, which affected three patients (12.5%), who required ICU admission.

In Figure 2, underlying diseases found on admission are presented. Ten patients (41.7%) had diabetes mellitus. Liver cirrhosis was found in seven patients (29.1%). An equal number (four) of patients (16.7%) had chronic hepatitis B and C. A small number of patients (8.3%) were diagnosed with Kaposi sarcoma, stroke, or cerebral toxoplasmosis. Three patients (12.5%) had dilative cardiomyopathy. Diabetes mellitus and liver cirrhosis were the most frequent comorbidities in our study group.

No statistical difference was detected in the baseline laboratory profiles between survivors and nonsurvivors (*p*-value > 0.05). A similar pattern was found with most CSF parameters, showing insignificant results between study groups, although the CSF opening pressure was significantly higher in nonsurvivors (28.3 vs. 16.3; *p*-value = 0.025, OR = 2.9, CI: [0.7–3.3]). In all patients with high CSF opening pressure, multiple lumbar punctures were performed during admission until normal values were obtained. Mortality was higher in patients with high opening pressure at admission. No significant difference was found in signs and symptoms between the studied groups, except for seizures at disease onset and ARDS, which had a significant effect on mortality (*p*-value = 0.001, OR = 1.4, CI: 1.1–1.9 and *p*-value = 0.001, OR = 1.8, CI: [1.2–2.2], respectively). Furthermore, the mortality rate was found to be significantly higher (*p*-value < 0.05) in patients who presented with fever, nausea/vomiting, visual impairment, altered sensorium, stiff neck, marked fatigue, diarrhea, upper extremity weakness, weight loss, generalized lymphadenopathy, and seizure. However, no independent risk factors for mortality were identified after running a multivariate analysis. The status of comorbid conditions found that only diabetes mellitus (*p*-value = 0.039, OR = 1.2, CI: [1.1–1.6]) was a significant risk factor for mortality in patients with cryptococcal meningitis. No significant association was found between mortality status and in-hospital complications (*p*-value > 0.05). Although there was no statistical significance in median hospital stay, we observed that the duration of stay was longer among the patients who died compared to those who survived (32 vs. 19 days). The median survival time of the patients treated in the hospital with cryptococcal meningitis secondary to AIDS was 40 days (CI: [28.8–51.2]).

## 4. Discussion

This study was conducted over ten years, finding 24 instances of CM diagnosis in our center. In Romania, on 31 December 2020, 16,848 people infected with HIV/AIDS were alive, out of a total of 25,486 since 1985. Additionally, many long-term survivors (>5500) from the age group 30 to 34 years coming from the cohort of 1987–1990 were alive. The number of people under treatment and postexposure prevention at the end of 2020, from the data recorded by the Support Unit Technical and Management (UATM), was 13,437 [16].

Our research revealed a higher proportion of men, which is consistent with the review conducted by Guess et al. [17], describing a discrepancy between males and females suffering from cryptococcosis. The main factors involved could be the higher number of male HIV patients, lower adherence to treatment, and increased exposure. Prominent symptoms described in our patients were nausea and vomiting (90.9%), headache (81.8%), altered sensorium (81.8%), visual impairment (72.7%), and fever (72.7%). Rarely, patients presented with a triad of fatigue, night sweats, and dyspnea (9.1%). In a retrospective study that included 1863 HIV-infected patients, Lakshmi et al. found that the most common signs and symptoms were headache (92.31%) and fever (79.49%) [18]. Kumar et al. noted headache, altered sensorium, and fever in all cases (100%) [19].

Research data, along with our study, describe headaches associated with increased intracranial pressure as a significant finding in patients with cryptococcal meningitis. Our patients underwent repeated lumbar punctures during their hospital stay, where elevated opening pressure correlated with higher morbidity and mortality. Similar findings were described by Brizendine et al. [20], where cryptococcemia and baseline opening pressure higher than 24 cm H_2_O were associated with increased odds of mortality.

Clinical manifestations were not specific for cryptococcal meningitis. In this study, seizures at onset had a significant effect on mortality. Furthermore, the mortality rate was found to be higher in patients who presented with fever, nausea/vomiting, visual impairment, altered sensorium, and marked fatigue. In a study conducted by Brizendine et al. [20], demographic and clinical features negatively associated with 90-day mortality were age < 50, presenting with headache or cough, and pulmonary site of infection. In our study, no patients had pulmonary cryptococcosis to the best of our knowledge, but similarly, a young age and headache were associated with higher mortality. Interestingly, the survival rate was higher in older patients than in younger patients, and the association was significant.

Our patients had severely altered CD4 counts, with average values lower than 25/mm^3^. In a study conducted by Singh et al. [21], fluconazole (200 mg thrice weekly) given to HIV-infected patients with CD4 cell counts of lower than or equal to 100/mm^3^ was efficacious as primary prophylaxis for cryptococcosis, with notably lower costs and increased convenience for patients in comparison with daily administration of the drug, in the pre-ART era. After the HAART program became accessible worldwide with a significant reduction of mortality and morbidity in HIV patients, prophylactic therapy for CM became unclear, primarily because of fluconazole resistance and drug interactions [22,23]. Additionally, results from a study conducted by Sungkanuparph et al. [24] concluded that there was no difference in the occurrence of newly diagnosed cryptococcosis between patients with and without primary prophylaxis. None of our 24 patients came to regular visits or blood analysis and all had low adherence to the ARV treatment. None of the patients took a fluconazole prophylaxys for CM. After discharge, all patients included in the study received oral fluconazole (400 mg/day for ten weeks) as consolidation therapy because of previously low adherence to the ART treatment. Of 24 patients, only one had a relapse.

The profile included low glucose levels, high protein, and elevated lactate levels regarding CSF laboratory findings. A study performed by Huang et al. [25] found that the baseline CSF protein concentrations were higher in patients with good outcomes. No statistical difference was detected in the baseline laboratory profile of CSF values and mortality. The antigen detection test for cryptococcal meningitis from CSF was positive in 100% of patients in this study. The World Health Organization recommends a cryptococcal antigen detection test as a screening tool for patients living with HIV. Rajasingham et al. concluded that CrAg screening is cost-effective and lifesaving for patients with HIV, with 77% of deaths being preventable [26].

The previously reported mortality rate for such patients was around 7–15% in Europe and North America [27]. Even in the era of the HAART program and access to antifungal therapy, the burden is still high. Our study found that high CSF opening pressure, seizures at onset of CM, liver cirrhosis, and decompensated liver cirrhosis represent negative predictors for mortality. In contrast to these findings, a study published by Bicanic et al. [8] concluded that raised CSF opening pressure did not influence the outcome, with the possible explanation that repeated serial lumbar punctures were performed according to local protocol. Graybill and colleagues found that patients with a CSF opening pressure greater than or equal to 25 cm H_2_O had higher CSF cryptococcal antigen titers [28]. One limitation of our study is that antigen titer was not available in the patients’ electronic chart. As this was retrospective research and used secondary data sources, there are, as usual, missing data and certain misclassifications.

## 5. Conclusions

Our findings conclude that cryptococcal meningitis occurs in most instances in long-standing HIV and cases of noncompliance with therapeutic orders for AIDS, when severely decreased CD4 levels allow most opportunistic infections to develop. The suspicion should remain high for these categories when presenting with slowly progressing signs of meningeal irritation, weight loss and fatigue, since rapid initiation of induction therapy with amphotericin B and flucytosine is life saving for two thirds of patients. The nonsurvivors are those who develop seizures at disease onset, high intracranial pressure, and ARDS, while diabetes mellitus is a significant predictor for disease severity.

## Figures and Tables

**Figure 1 diagnostics-12-00054-f001:**
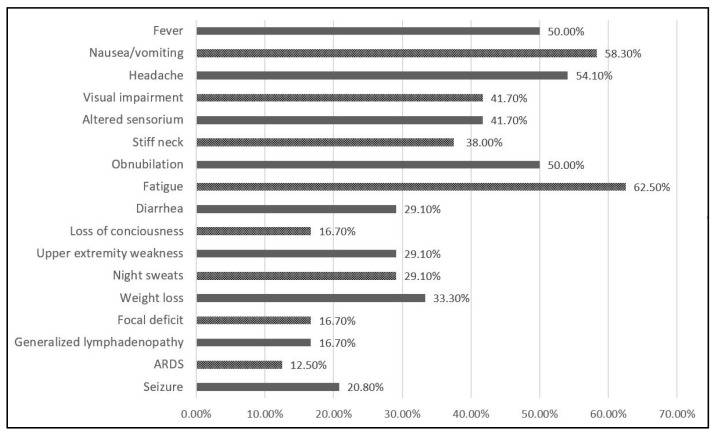
Sign and symptoms of patients with cryptococcal meningitis and immunodeficiency syndrome (AIDS).

**Figure 2 diagnostics-12-00054-f002:**
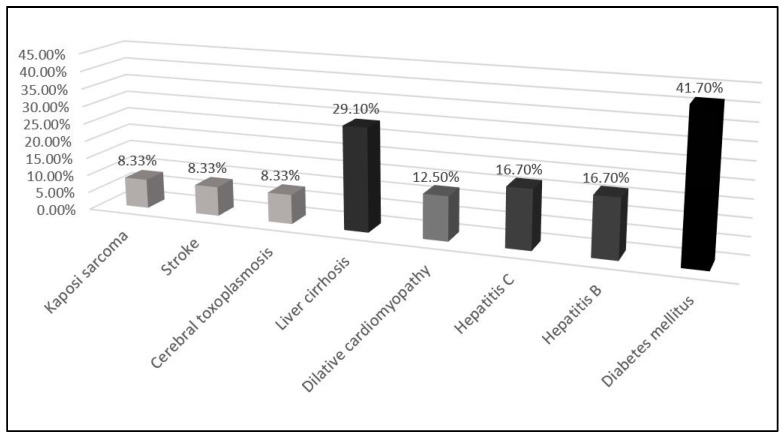
Underlying diseases in patients with cryptococcal meningitis and immunodeficiency syndrome (AIDS).

**Table 1 diagnostics-12-00054-t001:** Different approaches and therapies used for the management of *C. neoformans* meningitis.

AIDS Antiretroviral Treatment Regimens	Meningitis Treatment
**Regimen 1:** abacavir 600 mg + lamivudin 300 mg once a day	Amphotericin B (0.5–1 mg/kg/day) and oral flucytosine (150 mg/kg/day) for 2 weeks as induction therapy, followed by oral fluconazole (400 mg/day) for 10 weeks treatment, mannitol solution 20% (3 × 250 mL/day), and dexamethasone (4 mg/mL, 4 × 4 mg/day)
**Regimen 2:** lopinavir, ritonavir (Kaletra) 200 mg/50 mg; tabs, 800/200 mg (four 200/50 mg tabs) once daily and lamivudine 150 mg, zidovudine 300 mg; twice daily	
**Regimen 3:** lopinavir, ritonavir (Kaletra) 200 mg/50 mg; tabs, 800/200 mg (four 200/50 mg tabs) once daily, abacavir (Ziagen) tablets (300 mg) twice daily, lamivudine (Epivir) 300 mg tablets, twice daily	
**Regimen 4:** lopinavir, ritonavir (Kaletra) 200 mg/50 mg; tabs, 800/200 mg (four 200/50 mg tabs) once daily, abacavir 600 mg + lamivudine 300 mg (Kivexa) once daily	

**Table 2 diagnostics-12-00054-t002:** Demographics and clinical profile of patients.

Patient Characteristics	*n* (Frequency)	Mean ± SD (Range)
**Age**		41.36 ± 6.7 (32–52)
<40 years	10 (41.7%)	
≥40	14 (58.3%)	
**Gender**		
Female	7 (29.2%)	42.33 ± 8.5 (36–52)
Male	17 (70.8%)	41 ± 6.6 (32–51)
**Area of residence**		
Urban	6 (25.0%)	
Rural	18 (75.0%)	
**Etiology of HIV**		
Blood transfusion	5 (20.8%)	
Sexual contact	19 (79.2%)	
HIV Stage—AIDS	24 (100%)	
Serotype A *C.neoformans*	24 (100%)	
HIV viral load (thousand copies/µL)		145 ± 101 (42–1356)
**AIDS antiretroviral treatment regimen ***		
Regimen 1	1 (4.2%)	
Regimen 2	4 (16.7%)	
Regimen 3	6 (25.0%)	
Regimen 4	13 (54.1%)	
ICU admission	6 (25.0%)	
Duration of hospitalization (days)		25.27 ± 11 (14–42)
**In-hospital complications**		
Acute kidney injury	3 (12.5%)	
ARDS	3 (12.5%)	
Hepatocytolisis syndrome	2 (8.3%)	
Anemia	3 (12.5%)	
Septic shock	2 (8.3%)	
Oral candidiasis	4 (16.7%)	
Tuberculosis	4 (16.7%)	
Nosocomial pneumonia	2 (8.3%)	
Mortality	8 (33.3%)	

* As specified in Table 1.

**Table 3 diagnostics-12-00054-t003:** Laboratory findings.

	Variable	Value, Mean ± SD (Range)
**Serum Profile**	WBC (thousands/mm^3^)	12.5 ± 4.3 (9–24)
	Lymphocytes (%)	13.4 ± 16.9 (4–49)
	ESR (mm/h)	105.7 ± 15.3 (84–131)
	CRP (mg/L)	92.2 ± 116.3 (29–428)
	ASTO (<200 Todd units/mL)	121.1 ± 38.2 (80–200)
	Fibrinogen (g/L)	286.4 ± 71.3 (210–389)
	Ht (%)	26.8 ± 7.1 (11–39)
	Hg (g/dL)	9.2 ± 2.2 (8–13)
	AST (U/L)	191.6 ± 228.4 (12–642)
	ALT (U/L)	217.7 ± 275.9 (9–752)
	Creatinine (µmol/L)	0.60 ± 0.23 (0.22–0.90)
**CSF Profile**	WBC (thousands/mm^3^)	19.2 ± 7.9 (1–30)
	Protein (g/dL)	46.2 ± 47.4 (0.48–118.9)
	Glucose (mmol/L)	2.4 ± 0.95 (1.2–4.2)
	Lactate (mmol/L)	19.2 ± 13.1 (3.8–54.6)
	Opening pressure (cm H_2_O)	21.1 ± 8.8 (8–29)
	CD4 (cells/µL)	24.9 ± 22.1 (9–69)
	CD8 (cells/µL)	162.5 ± 349.3 (29–1215)
	CD3 (cells/µL)	62.6 ± 37.2 (289–154)
	CD4/CD8 (cells/µL)	0.36 ± 0.5 (0–1)

WBC—white blood cells; ESR—erythrocyte sedimentation rate; CRPC—reactive protein; ASTO—antistreptolysin O; Ht—hematocrit; Hg—hemoglobin; AST—aspartate aminotransferase; ALT—alanine aminotransferase.

**Table 4 diagnostics-12-00054-t004:** Association of study variables with patient outcomes.

Study Variables		Outcome		*p*-Value
		Died (*n* = 6)	Survived (*n* = 18)	
**Age group**	<40 years (*n* = 11)	5 (45.5%)	6 (54.5%)	0.033 ^₣,^*
	≥40 years (*n* = 13)	1 (7.7%)	12 (92.3%)	
**Gender**	Female (*n* = 7)	2 (28.5%)	5 (71.5%)	0.795 ^₣^
	Male (*n* = 17)	4 (23.5%)	13 (76.5%)	
**Area of residence**	Urban (*n* = 6)	3 (50.0%)	3 (50.0%)	0.102 ^₣^
	Rural (*n* = 18)	3 (16.6%)	15 (83.4%)	
**HIV transmission**	Blood transfusion (*n* = 5)	2 (40.0%)	3 (60.0%)	0.383 ^₣^
	Sexual contact (*n* = 19)	4 (21.0%)	15 (79.0%)	
**HIV viral load** (thousand copies/µL)		158 ± 87	139 ± 114	0.713 ^t^
**AIDS antiretroviral treatment regimen**	Regimen 1 (*n* = 1)	1 (100%)	0 (0.0%)	0.276 ^χ^
	Regimen 2 (*n* = 4)	1 (25.0%)	3 (75.0%)	
	Regimen 3 (*n* = 6)	2 (33.3%)	4 (66.7%)	
	Regimen 4 (*n* = 13)	2 (15.4%)	11 (84.6%)	
**Serum Profile**	WBC (thousands/mm^3^)	11.5 (10.3–21)	10 (9–14)	0.444 ^Ư^
	Lymphocytes (%)	4.5 (4–6)	8 (5–46)	0.065 ^Ư^
	ESR (mm/h)	111 ± 11.6	102.7 ± 17.1	0.415 ^t^
	CRP (mg/L)	79.5 (40.3–344.5)	35 (34–100)	0.449 ^Ư^
	ASTO (<200 Todd units/mL)	137.5 ± 53.2	111.7 ± 27.1	0.305 ^t^
	Fibrinogen (g/L)	283 ± 71.9	288.3 ± 76.6	0.913 ^t^
	Hematocrit (%)	29.5 (15.5–33)	26 (23–29)	0.510 ^Ư^
	Hemoglobin (g/dL)	11 (9.3–12.8)	9 (8–13)	0.565 ^Ư^
	AST (U/L)	109 (15.3–205.8)	94 (25–598)	0.394 ^Ư^
	ALT (U/L)	88.5 (14.8–181)	149 (32–741)	0.298 ^Ư^
	Creatinine (µmol/L)	0.65 ± 0.32	0.58 ± 0.19	0.665 ^t^
**CSF Profile**	WBC (thousands/mm^3^)	18.8 ± 12.7	19.4 ± 4.9	0.900 ^t^
	Protein (g/dL)	29.2 (3.4–98.5)	23.8 (14.2–117.9)	0.450 ^Ư^
	Glucose (mmol/L)	2.3 ± 0.78	2.5 ± 1.08	0.785 ^t^
	Lactate (mmol/L)	13.9 (5.5–18.6)	20.9 (14.3–21)	0.088 ^Ư^
	Opening pressure (cm H_2_O)	28.3 ± 0.96	16.3 ± 8.4	0.025 ^t,^*
	CD4 (cel/µL)	33 ± 31.6	20.3 ± 15.6	0.386 ^t^
	CD8 (cel/µL)	68.5 (53.8–68.5)	59 (46–62)	0.256 ^Ư^
	CD3 (cel/µL)	63.5 (31.5)	49 (34–73)	0.850 ^Ư^
	CD4/CD8 (cel/µL)	0.53 (0.09–0.97)	0.10 (0.01–0.26)	0.053 ^Ư^
**Sign and symptoms**	Fever	6 (100%)	6 (33.3%)	0.004 ^₣,^*
	Nausea/vomiting	6 (100%)	8 (44.4%)	0.029 ^₣,^*
	Headache	5 (83.3%)	8 (44.4%)	0.156 ^₣^
	Visual Impairment	6 (100%)	4 (22.2%)	0.001 ^₣,^*
	Altered sensorium	5 (83.3%)	5 (27.8%)	0.017 ^₣,^*
	Stiff neck	5 (83.3%)	4 (22.2%)	0.013 ^₣,^*
	Fatigue	6 (100%)	8 (44.4%)	0.029 ^₣,^*
	Diarrhea	4 (66.7%)	3 (16.7%)	0.019 ^₣,^*
	Loss of consciousness	2 (33.3%)	2 (11.1%)	0.205 ^₣,^*
	Upper extremity weakness	4 (66.7%)	3 (16.7%)	0.019 ^₣,^*
	Night sweats	3 (50.0%)	4 (22.2%)	0.194 ^₣,^*
	Weight loss	5 (83.3%)	3 (16.7%)	0.002 ^₣,^*
	Focal deficit	2 (33.3%)	2 (11.1%)	0.205 ^₣,^*
	Generalized lymphadenopathy	3 (50.0%)	1 (5.6%)	0.011 ^₣,^*
	Seizure	4 (66.7%)	1 (5.6%)	0.001 ^₣,^*
**Comorbidities**	Kaposi sarcoma	2 (33.3%)	0 (0%)	0.010 ^₣,^*
	Stroke	1 (16.7%)	1 (5.6%)	0.393 ^₣^
	Cerebral toxoplasmosis	2 (33.3%)	0 (0%)	0.010 ^₣,^*
	Liver cirrhosis	4 (66.7%)	3 (16.7%)	0.019 ^₣,^*
	Dilated cardiomyopathy	2 (33.3%)	1 (5.6%)	0.074 ^₣^
	Hepatitis C	3 (50.0%)	1 (5.6%)	0.011 ^₣,^*
	Hepatitis B	2 (33.3%)	2 (11.1%)	0.205 ^₣^
	Altered mental status	1 (16.7%)	1 (5.6%)	0.393 ^₣^
	Diabetes Mellitus	6 (100%)	4 (22.2%)	0.001 ^₣,^*
**Admitted to ICU**		2 (33.3%)	1 (5.6%)	0.074 ^₣^
**In-hospital complication**	Hepatocytolisis syndrome	1 (16.7%)	1 (5.6%)	0.393 ^₣^
	Acute kidney injury	1 (16.7%)	1 (5.6%)	0.393 ^₣^
	Anemia	1 (16.7%)	2 (11.1%)	0.721 ^₣^
	Septic shock	3 (50.0%)	0 (0%)	0.001 ^₣^
	Oral candidiasis	2 (33.3%)	1 (5.6%)	0.074 ^₣^
	Tuberculosis	2 (33.3%)	1 (5.6%)	0.074 ^₣^
	Nosocomial pneumonia	1 (16.7%)	0 (0%)	0.076 ^₣^
	ARDS	3 (50.0%)	0 (0.0%)	0.001 ^₣^
**Hospital stay (days)**		32 (17.3–41.5)	19 (16–32)	0.296 ^Ư^

Frequency (%); mean ± standard deviation; median (interquartile range); Fisher exact test = ₣; independent *t*-test = t; Mann–Whitney U test = Ư, chi-square test = χ; * significant values at α = 0.05.

## Data Availability

Data available on request.

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
