# Peer review of "Clinical Profile of 24 AIDS Patients with Cryptococcal Meningitis in the HAART Era: A Report from an Infectious Diseases Tertiary Hospital in Western Romania"

_diagnostics, 2021, doi:10.3390/diagnostics12010054_

Round 1
Reviewer 1 Report
line 115 - All patients with ICD-10 diagnostic code B45.1 (cerebral Cryptococcosis),- irrelevant, please delete
line 127 - with the registration number HIDPVB2154-irrelevant, please delete
Table 1, Table 2 - Meningitis treatment 1,2,3,4 - the antifungal treatment is identical in all four groups. I don't think antibiotic treatment is important, please correct, I suggest keeping one type of treatment, the antifungal one
Table 1 - ASLO value- irrelevant for study
line 205-209 - please delete, I do not think there is a link between the mortality rate and the treatment regimen used, all patients were viral failure
Table 3 - suggest deleting AIDS antiretroviral treatment and Meningitis treatment
References - suggest adding
Streinu-Cercel A, Săndulescu O, Poiană C, Dorobanțu M, Mircescu G, Lăzureanu VE, Dumitru IM, Chirilă O, Streinu-Cercel A, Extended Consensus Group. Consensus statement on the assessment of comorbidities in people living with HIV in Romania. GERMS 2019;9(4):198-210. doi: 10.18683/germs.2019.1178.
Author Response
Dear reviewer,
We greatly appreciate your assistance in helping us improve our manuscript. Please follow the following changes based on the reviewer’s feedback:
- Line 115 (now 116): We deleted lines 116 and 118, as they were considered irrelevant and repetitive.
- Line 127 (now 128): We deleted line 128, with the registration number.
- Table 1 and Table 2: We removed the lines describing 4 different meningitis treatment types, as they mostly consisted in the same induction and maintenance scheme with Amp B+Flucytosine and Fluconazole, while the only significant difference was an additional antibiotic for bacterial prophylaxis that is unrelated to fungal infections.
- Table 1 and line 169: We removed the lines mentioning ASLO analysis, as it is irrelevant for the study.
- Lines 205-209 were deleted, as there was no link between the mortality rate and the treatment regimen used, since meningitis treatment was the same, and all patients were uncompliant to AIDS treatment.
- Table 3: We considered deleting the meningitis treatment lines, as mentioned at the bullet line number 5.
- The reference “Streinu-Cercel A, Săndulescu O, Poiană C, DorobanÈ›u M, Mircescu G, Lăzureanu VE, Dumitru IM, Chirilă O, Streinu-Cercel A, Extended Consensus Group. Consensus statement on the assessment of comorbidities in people living with HIV in Romania. GERMS 2019;9(4):198-210. doi: 10.18683/germs.2019.1178” was added at number 15.
It is also important to mention that the manuscript underwent extensive corrections using the MDPI English editing service, while other corrections and edits were performed based on given feedback from reviewers.
Best regards,
The authors
Reviewer 2 Report
The article complies with the requirements for writing a scientific paper. The importance of the topic lies in the follow-up of the clinical-evolutionary and treatment characteristics of Cryptococcus neoformans meningitis in HIV-positive patients and the challenges regarding the diagnosis and management of these patients. In addition, I believe that the article deserves to be published as a reference for other studies and comparisons with samples of patients from different parts of the world.
Author Response
Dear reviewer,
We greatly appreciate your positive feedback. Please consider the following adjustments that were performed based on other comments:
- The manuscript underwent extensive corrections using the MDPI English editing service.
- Line 115 (now 116): We deleted lines 116 and 118, as they were considered irrelevant and repetitive.
- Line 127 (now 128): We deleted line 128, with the registration number.
- Line 135: Added a new reference in the methodology section, with the number 15.
- Table 1 and Table 2: We removed the lines describing 4 different meningitis treatment types, as they mostly consisted in the same induction and maintenance scheme with Amp B+Flucytosine and Fluconazole, while the only significant difference was an additional antibiotic for bacterial prophylaxis that is unrelated to fungal infections.
- Line 169 and Table 1: We removed the lines mentioning ASLO analysis, as it is irrelevant for the study.
- Lines 207-209 were deleted, as there was no link between the mortality rate and the treatment regimen used, since meningitis treatment was the same, and all patients were uncompliant to AIDS treatment.
- Table 3: We considered deleting the meningitis treatment lines, as mentioned at the bullet line number 5.
- Line 298: We corrected the number of patients diagnosed (24).
- We corrected the figures in order to comply with the publisher’s requirements.
- To separate laboratory findings from clinical course we divided table 1 to two separate tables.
- In table 3 (not table 4), the statistical analysis was performed using Fisher’s exact test for small sample size (where p-value is marked with an F), instead of Chi-square.
- The HIV transmission in table 4 is mentioned as a factor for comparing proportions of deceased and survivors.
- Regarding table layout: we are waiting for editorial advice, if we should keep them as they are, or remove inside horizontal lines with the risk of creating confusion when reading the tables.
- Line 282 (now 266): we corrected the first lines to follow the main arguments of the paragraph.
- We removed the parts repeating statistical results in the discussion section.
- Lines 334-338 in the Conclusions section were deleted, as considered irrelevant for this section.
Best regards,
The authors
Reviewer 3 Report
The paper aims to describe 24 patients with AIDS and cryptococcal infection of the CNS. The interesting about the article is that it comes from a country with some special problems. Unfortunately the manuscript has many flaws which must be corrected in order to be published. English language needs also significant amelioration.
Introduction.
The introduction is too long. All this information about cryptococcus (3-4 paragraphs) is unnecessary.
Furthermore, you are including results and discussion in the last paragraph. This has no place there.
Results.
Table 1.
You must separate laboratory findings from clinical course. Divide table 1 to two separate tables.
There are several problems about the way you are presenting the results in the table.
ASTO , not ASLO.
In the results you must not repeat what you are already showing in the tables.
In table 3. The number of patients is too small to make statistical analysis.
In table 3. You have included HIV transmission as a factor for survival. How could you explain that?
In tables , lines must not be shown.
Discussion.
Why are you talking about 11 patients (first line) while you have already said that you have 24? ?????
In the second paragraph you are repeating the results. This must not be done except if you are comparing them with other papers.
Line 282. First line. What is the relation of the first sentence with the rest of the paragraph?
Conclusions. Conclusions are conclusions. Results again should not be mentioned. Or what they received for treatment.

Author Response
Dear reviewer,
We greatly appreciate your assistance in helping us improve our manuscript. Please follow the following changes based on the reviewer’s feedback:
- The manuscript underwent extensive corrections using the MDPI English editing service.
Introduction
- In order to shorten the introduction part, we deleted one paragraph.
- We deleted a sentence from the last paragraph that included results (line number 103).
Results
- To separate laboratory findings from clinical course we divided table 1 to two separate tables.
- We removed the lines mentioning ASLO analysis, as it is irrelevant for the study.
- We tried to delete the parts repeating the same information as in the tables.
- In table 3 (now table 4), the statistical analysis was performed using Fisher’s exact test for small sample size (where p-value is marked with an F), instead of Chi-square.
- The HIV transmission in table 3 (now table 4) is mentioned as a factor for comparing proportions of deceased and survivors.
- Regarding table layout: we are waiting for editorial advice, if we should keep them as they are, or remove inside horizontal lines with the risk of creating confusion when reading the tables.
Discussion
- Line 302: There was a typo, and we corrected the number of patients diagnosed (24).
- We modified the second paragraph to avoid repeating statistical results from the previous section.
- Line 282 (now 266): we corrected the first lines to follow the main arguments of the paragraph.
Conclusions
- Lines 334-338 were deleted, as considered irrelevant for this section.
Best regards,
The authors
Round 2
Reviewer 3 Report
Dear authors you have made several progress. Nevertheless first your English needs further amelioration. Second I have made several remarks about the content. Third I would like to have some lines about what is your manuscript is adding to scientific knowledge.
Finally in the conclusion, you say that few data are available on cryptococcal meningitis". By seraching in Pubmed : Cryptococcal meningitis in HIV, I found 1699 results, and some very recent interesting papers. Therefore, I would like another conclusion as well as the use of recent literature in the discussion, apart from the other remarks.

Author Response
Dear Reviewer,
Thank you for taking time to closely analyse our article. We believe your feedback was valuable, so, we made all possible changes based on your recommendations along with the Academic Editor’s notes. We used the “Track Changes” option for all edits, that can be found in the attached document, and explained below:
- Several important changes were required for Table 1:
- Since Table 1 (now Table 2) was too difficult to edit, we decided to create a new table that includes the same information but is easier to read.
- The unit of measurement (thousand copies/µL) for HIV viral load was added in the table and line 211 (previously line 202).
- AIDS treatment 1,2,3, and 4 are described now in Table 1. We added the word “regimen” to all rows reporting AIDS treatment.
- Unfortunately, we do not have access to the data whether sexual intercourse was heterosexual or homosexual since patient records do not hold this information. Moreover, patients are often reluctant to seek medical care until severe complications occur, yet many refuse to say the truth or to disclose such information. We decided to add this information in text at lines 149-153.
- The order of tables was changed.
- We were meaning stroke and altered mental status instead of hemiparesis and intellectual disability, respectively. These terms were replaced now, both in text and figures/tables.
- Corrections were made to all values in Table 4.
- Several changes were done to Table 3 and Table 4, including the unit of measurement to all quantitative variables.
- We revised the references, excluding the old literature and replacing with more recent studies.
- The conclusion part was entirely remade to clarify our aims and findings that are important for the reader.
Best regards,
The authors
